# Nrf2 and Antioxidant Response in Animal Models of Type 2 Diabetes

**DOI:** 10.3390/ijms24043082

**Published:** 2023-02-04

**Authors:** R. Paul Robertson

**Affiliations:** Division of Metabolism, Endocrinology and Nutrition, Department of Internal Medicine, University of Washington, Seattle, WA 98105, USA; rpr@uw.edu

**Keywords:** Nrf2, antioxidant response, type 2 diabetes in animals

## Abstract

This perspective examines the proposition that chronically elevated blood glucose levels caused by type 2 diabetes (T2D) harm body tissues by locally generating reactive oxygen species (ROS). A feed-forward scenario is described in which the initial onset of defective beta cell function T2D becomes sustained and causes chronic elevations in blood glucose, which flood metabolic pathways throughout the body, giving rise to abnormally high local levels of ROS. Most cells can defend themselves via a full complement of antioxidant enzymes that are activated by ROS. However, the beta cell itself does not contain catalase or glutathione peroxidases and thereby runs a greater risk of ROS-induced damage. In this review, previously published experiments are revisited to examine the concept that chronic hyperglycemia can lead to oxidative stress in the beta cell, how this relates to the absence of beta cell glutathione peroxidase (GPx) activity, and whether this deficiency might be ameliorated by genetic enrichment of beta cell GPx and by oral antioxidants, including ebselen, a GPx mimetic.

## 1. Introduction

Much more is known about the clinical phenotype of type 2 diabetes (T2D) than is known about its primary genetic causes. Accordingly, it is accurate to consider T2D as a syndrome of polygenic origin rather than a single genetic illness. The prevalence and incidence of diabetes in the world continue to climb to unprecedented heights, so it is important to consider the possible oxidative features of the internal and external environments that impact mammals and humans who have developed T2D. Two important variables are the oxidative stress that can be caused by chronic hyperglycemia and the competency of the consequent innate antioxidant response. In this perspective, I revisit experiments designed to evaluate the success of endogenous Nrf2 and antioxidant gene expression to repair damaged beta cells in T2D.

## 2. Background: The Pancreatic Islet as a Vulnerable Target for Reactive Oxidative Species

Islets comprise approximately 2–3% of the pancreatic mass and consist primarily of beta cells and alpha cells, amongst others. Beta cells synthesize and secrete insulin in response to rising blood glucose levels, whereas alpha cells synthesize and secrete glucagon in response to hypoglycemia. The primary defect in T2D is defective glucose-induced insulin secretion, which is made worse in the presence of general insulin resistance, a condition that frequently accompanies the development of the syndrome of T2D, obesity, and hyperlipidemia.

Physiologic levels of ROS from various sources contribute positively to many cellular processes. However, chronic hyperglycemia can lead to excessive formation of ROS via several metabolic pathways, including oxidative phosphorylation, glyceraldehyde autoxidation, PKC activation, and metabolism of hexosamine, glucosamine, dihydroxyacetone, enediol, and glyceraldehyde ([1,2]; Figure 1). If the antioxidant regulatory response at the tissue level is not sufficient to neutralize excessive ROS, structural tissue damage can ensue, which is referred to as oxidative stress. Most organs have tissues with cells that contain a full complement of antioxidants, including catalase, superoxide dismutases, glutathione peroxidases, and hemoxygenases. Pancreatic beta cells are different and thus particularly at risk for oxidative stress because they are unique among other tissues in having only partial protection with superoxide dismutases and hemoxygenases [3,4,5,6,7,8,9,10,11,12,13,14,15]. Thus, under conditions of chronic hyperglycemia, which leads to a general state of increased formation of ROS, the beta cell is especially at risk for undergoing the toxic effects of abnormal amounts of ROS. This means that the beta sell, whose innate function is abnormal in T2D, can be both the cause and a secondary victim of the chronic hyperglycemia it causes ([16,17,18,19,20]; Figure 2). Many brief clinical trials with traditional oral antioxidants to treat the hyperglycemia of T2D have been attempted but have not yet been shown to be effective in humans. More recent experimental data suggest that the innate beta cell Nrf2/antioxidant pathway in beta cells is active in self-repair early after exposure to oxidative stress caused by a high-fat diet (see Figure 3, Figure 4 and Figure 5 below), which raises the question whether this phenomenon might be clinically applicable to the treatment of T2D.

## 3. Molecular Consequences of Chronic Hyperglycemia and Resultant Oxidative Stress on Residual Beta Cell Function in T2D

In the last half of the 20th century, a great deal was learned about abnormalities in beta cell function in humans and animals with T2D. Much of this information came from clinical research that involved studies such as the oral and intravenous glucose tolerance tests (for a general review, the reader is referred to DeGroot Endocrinology, 8th edition, year 2023, pages 557–588). T2D was characterized metabolically as a state involving not only hyperglycemia but paradoxically also one in which fasting circulating insulin levels could be in the normal or even in the distinctly abnormally high range. Over several decades it became appreciated that high basal insulin levels were common in T2D and were, for the most part, associated with individuals who were obese and, to a lesser extent, lean. It was also learned from intravenous glucose tolerance testing that the initial burst of insulin release within minutes of injecting glucose that was normally present in non-diabetic humans was totally absent in subjects with T2D. This loss was unique to glucose because the first-phase responses to other agonists, such as amino acids, glucagon, and isoproterenol, were fully intact. These features strongly suggested that T2D was clearly distinct from T1D, an immunologic disease that involved the death of beta cells and the total disappearance of agonist-induced insulin responses.

In roughly the same time frame, there was intense research attention on exploring the causes of elevations in blood insulin levels in obese individuals with and without T2D. The primary in vivo methodology initially involved glucose clamp studies, which can be performed under normoglycemic or hyperglycemic conditions. The overall study design has several different variations, but the overall goal is to ascertain the quantitative relationship between blood glucose and insulin levels as one or the other is manipulated independently. For one example, it was observed that obese and aged individuals with elevated basal insulin levels were resistant to the effects of exogenous insulin such that more infused insulin was required to maintain a given clamped glucose value. Such findings gave rise to the term insulin resistance and led to a great deal of in vitro research examining molecular mechanisms of insulin action and insulin resistance in various tissues, including the liver, muscle, and adipose tissue.

Research into the regulation of beta cell function revealed that insulin gene expression in cell lines was adversely affected if the cells were chronically cultured in media containing high concentrations of glucose. For example, experiments using the beta cell line HIT-T15 showed that chronic culturing of these cells in media containing supraphysiologic concentrations over many weeks caused progressive declines in levels of insulin mRNA, insulin content, and glucose-induced insulin secretion [24]. Loss of agonist-induced insulin secretion in these cells was a general phenomenon rather than the glucose-specific one observed in T2D subjects [25]. In other words, type 1 patients have no insulin to secrete, whereas type 2 patients have intact insulin secretion but not in response to an acute intravenous glucose challenge. This distinction suggested that chronic exposure to high glucose levels per se could lead to adverse changes in beta cells independent of the innate nature of T2D. This, in turn, suggested that the hyperglycemia caused by T2D itself might have an independent and secondary consequence of further impairing the ability of beta cells to synthesize and secrete insulin. Searching for such a feed-forward deleterious mechanism that might worsen already compromised beta cell function led to the observations using HIT-T15 cells that prolonged exposure to supraphysiologic glucose concentrations caused the disappearance of two insulin promoter regulatory proteins. One protein was initially named GSTF [26] and later identified as STF-1 [27], and even later PDX-1. A second protein that disappeared was initially named RIPE3b1 activator [28,29] and later termed MafA. Studies of DNA binding to the insulin promoter of HIT-T15 cells showed that binding of MafA decreased 20 passages earlier than the decrease in Pdx-1 binding [30] and also that insulin reporter activity could be partially restored by reconstitution of the cells with Pdx-1 DNA. Results from similar experiments using another beta cell line, the Beta-TC-6 cell, demonstrated similar adverse effects of prolonged exposure to high glucose levels on insulin gene expression, but in this case, the defect was primarily a decrease in RIPE-3b1 activator with no defect in PDX-1 gene expression [29].

Extension of these in vitro experiments to in vivo experiments with a rodent model of T2D, the male Zucker Diabetic Rat (ZDF) fed a high-fat diet, were conducted by comparing data from control rats and rats treated with troglitazone, a drug that lowers blood glucose in these rodents. The control animals developed progressive hyperglycemia and an associated loss of PDx-1 and insulin mRNAs and diminution of glucose-stimulated insulin secretion. Treatment with troglitazone prevented hyperglycemia and the adverse effects on PDX-1 and insulin gene expression and improved insulin secretion [18].

## 4. Search for Causal Relationships among Glucose Toxicity, Oxidative Stress, and Defects in Beta Cell Function in T2D: The Importance of Glutathione Peroxidase

Clinical reports of elevated levels of oxidative stress markers in patients with T2D are numerous. Relevantly, the islet is among the least well-endowed tissues in terms of intrinsic antioxidant enzyme expression. Essentially, the beta cell expresses only superoxide dismutases and hemoxygenases with very little, if any, glutathione peroxidase or catalase [3,4,5,15], which leaves the cell largely unprotected against intracellular peroxides. Additionally, the product of superoxide dismutases is hydrogen peroxide, which itself a reactive oxygen species. To assess the impact of these inter-relationships, we cultured HIT-T15 cells in media containing supraphysiologic levels of glucose with and without the inclusion of the antioxidant N-acetyl-l-cysteine (NAC) or aminoguanidine (AG) in the media. Both NAC and AG partially prevented decreases in insulin mRNA, insulin gene promoter activity, DNA binding of PDX-1 and MafA, insulin content, and glucose-induced insulin secretion [16]. Additionally, these two drugs were given to hyperglycemic ZDF diabetic rats, both of which prevented a rise of blood markers of oxidative stress (8-hydroxy-2′-deoxyguanosine and malondialdehyde + 4-hydroxy-2-nonenal) and partially prevented hyperglycemia, glucose intolerance, and defective insulin secretion as well as decrements in cell insulin content, insulin gene expression and PDX-1 binding of the insulin gene promoter.

These observations led to an assessment of the importance of the absence of glutathione peroxidase activity in beta cells during periods of oxidative stress. These experiments involved both HIT-T15 cells and isolated islets from male Wistar rats and from humans [17]. We observed that high glucose concentrations increased intracellular peroxide levels in human islets and HIT-T15 cells. Inhibition of gamma-glutamyl cysteine synthetase by buthionine sulfoximine augmented the increase in islet peroxide and decrease in insulin mRNA caused by ribose. Adenoviral overexpression of glutathione peroxidase increased islet glutathione peroxidase activity and protected islets against the adverse effects of ribose. We concluded that oxidative stress is one mechanism for glucose toxicity in pancreatic islets, a mechanism that is enhanced by the innate absence of islet glutathione peroxidase. They suggested that the use of GPx mimetics may represent a valuable ancillary treatment that could add a novel layer of protection for the beta cell [18,31].

## 5. Effects of Ebselen, a Glutathione Mimetic, in Preventing T2D in the Male ZDF Rat

Ebselen (2-phenyl-1,2-benzisoselenazol-3(2H)-one, a nontoxic seleno-organic drug, is a lipid-soluble, orally bioavailable small molecule classified as a GPx mimetic [32,33,34] that has been used in human trials for hearing loss and neurovascular disease [35,36]. Because beta cells are known to be deficient in GPx, we selected ebselen as a candidate for preventing the deterioration of beta cells in male ZDF rats, a rodent model of T2D [22]. These animals are obese, leptin receptor-negative, and develop glucose intolerance, insulin resistance, and fasting hyperglycemia. Ebselen treatment of male ZDF rats with T2D was found to have beneficial effects on beta cell function and structure. It ameliorated fasting hyperglycemia, sustained non-fasting insulin levels, and lowered non-fasting glucose and HbA1c levels with no effects on body weight. It also doubled beta cell mass (Figure 3 and Figure 4), prevented apoptosis, prevented the expression of oxidative stress markers, and enhanced intranuclear localization of the two critical insulin transcription factors, PDX-1) and MafA. These findings in live animals suggest that ebselen is an agent that should be evaluated clinically in humans undergoing the onset of type 2 diabetes

## 6. Protection of Beta Cells by the GPx-1 Transgene against the Glucotoxic Effects of Chronic Hyperglycemia Female *db/db* Mice

We studied whether an increase in GPx-1 in beta cells specifically would protect them from the adverse effects of chronic hyperglycemia [21]. For these studies, we used female C57BLKS/J mice fed a high-fat diet, an animal model of T2D, to overexpress the GPx-1 transgene. We found that without the assistance of hypoglycemic agents, spontaneous hyperglycemia in *db/db-GPx*(+) mice was initially ameliorated by ten weeks of age and then by 20 weeks almost completely reversed (compare the red line in Figure 3 at 10 weeks vs. 20 weeks. Beta cell volume and insulin granulation and immunostaining were greater in *db/db*-GPx-1(+) overexpressed animals compared to *db/db*-GPx-1(−) control animals. The control animals lost intranuclear MafA, which was prevented by GPx-1 overexpression. We concluded that it was a transgenic enhancement of intrinsic antioxidant defenses of mouse beta cells that protected them against the deterioration of beta cells that occurs during the development of hyperglycemia. These results led us to examine more closely whether animals just beginning to develop spontaneous T2D and whose beta cells were just on the verge of being exposed to initial hyperglycemia and oxidative stress might evince evidence for intrinsic antioxidant expression via the Nrf2/antioxidant pathway as an attempt to initiate beta cell repair [23].

## 7. The Innate Protective Response of the Nrf2 and Hemoxygenase-1 Pathway against the Development of T2D Diabetes in Female Fat-fed ZFD Rats

The strategy of these studies was to examine the evidence for endogenous antioxidant-mediated repair mechanisms via activation of the Nrf2/antioxidant pathway ([37,38,39,40,41,42,43,44]; Figure 6) that might occur during early beta cell deterioration characteristic of female ZDF rats fed a high-fat diet. These studies involved feeding rats with a high-fat diet (HFD) for 1, 2, 4, 7, 9, 18, or 28 days followed by a return to regular chow for 2–3 weeks after stopping the HFD [23]. We observed evidence of functional beta cell damage (hyperglycemia) after 9 days of exposure to HFD. We also observed a return to normoglycemia after 2–3 weeks of stopping the HFD and returning the rats to normal chow. Damage was more severe, and repair was less evident after 18- or 28-day exposure to HFD. Assessment of beta cell volume, morphology, and insulin-specific immunoreactivity, as well as ultrastructural analysis by transmission electron microscopy, revealed that short-time exposure to HFD produced significant changes in morphology and function that were reversed after returning to regular chow for two weeks. Contemporaneous formation of intracellular markers of oxidative stress, intranuclear translocation of Nrf2, and formation of intracellular antioxidant proteins in the 9-day experiments indicated participation of the Nrf2/antioxidant pathway in this reversal (Figure 5). This repair did not occur if the animals were immediately sacrificed at 9 days of HFD without returning them to regular chow for two weeks before sacrifice.

## 8. Conclusions

The main emphasis of this perspective is that oxidative stress caused by chronic hyperglycemia is an important force in the continued deterioration of beta cell function in T2D and is relentless in the face of inadequately treated chronic hyperglycemia. When hyperglycemia becomes a constant feature of T2D, beta cell damage is amplified and sustained by its own failure to correct excessive levels of glucose and reactive oxygen species. This begets increasingly more beta cell damage, a vicious self-perpetuating cycle that represents a feed-forward mechanism for even more cellular damage. A caveat to this logic is that animal models of T2D also develop high levels of blood triglyceride, which raises the possibility lipids and not necessarily glucose, may play a role in the progressive demise of beta cell function, i.e., via lipotoxicity [45]. We assessed this possibility in experiments involving Zucker diabetic rats treated for 6 weeks with either bezafibrate, a lipid-lowering drug that does not affect plasma glucose levels, or with phlorizin, a drug that lowers plasma glucose without lowering lipid levels [46]. We observed that antecedent hyperglycemia, not hyperlipidemia, was associated with increased islet triacylglycerol content and decreased insulin gene expression and thus glucotoxicity and not lipotoxicity, was more likely to account for the demise of beta cell function in the ZDF animals.

The evidence cited in this perspective and brief review suggest that incomplete resolution of hyperglycemia characteristic of medical treatment of T2D is itself a factor in the continued development of beta cell deterioration over time and that chronic oxidative stress plays a major role in this pathogenesis. It seems possible that clinical strategies involving antioxidants should enhance protection from persistent hyperglycemia and that this might allow the completion of beta cell repair and resolution of hyperglycemia after the onset of the disease.

## Figures and Tables

**Figure 1 ijms-24-03082-f001:**
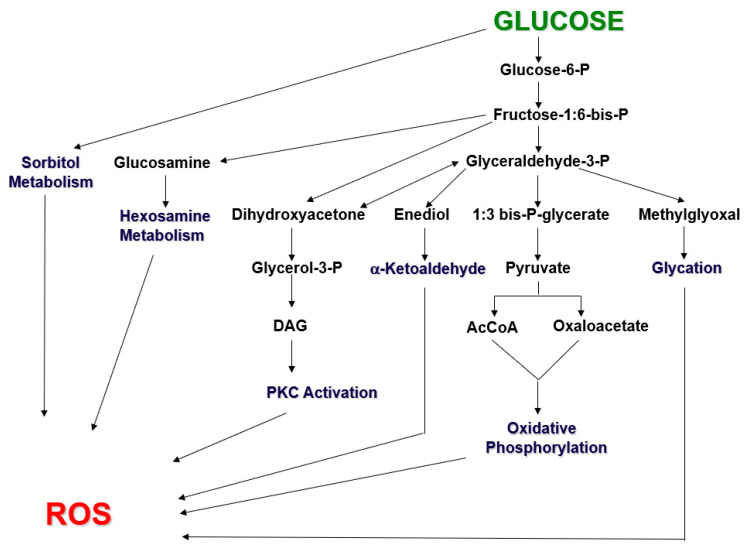
Metabolic pathways along which glucose metabolism can form reactive oxygen species (ROS). Under physiologic conditions, glucose primarily undergoes glycolysis and oxidative phosphorylation. Under hyperglycemic conditions, excessive glucose levels can swamp the glycolytic process and glyceraldehyde catabolism, so that metabolites are shunted to other pathways, which then generate increasing levels of ROS. Modified from reference [1].

**Figure 2 ijms-24-03082-f002:**
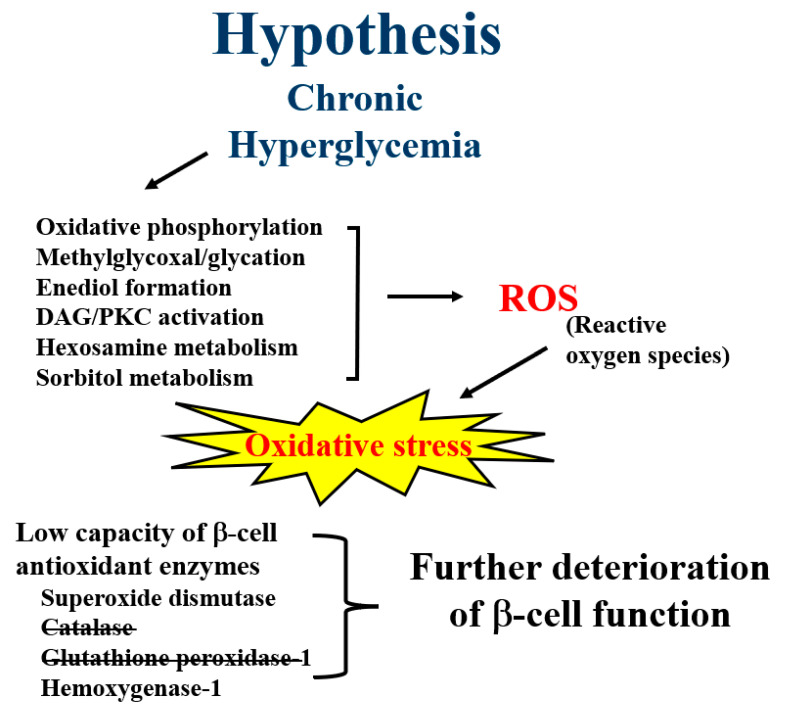
Hypothesis: Under conditions of supraphysiologic concentrations of blood glucose, various metabolic pathways generate increasing levels of ROS in body tissues, including the pancreatic beta cell. The normal beta cell, however, does not contain the full complement of antioxidant enzymes found in other tissues, specifically catalase and glutathione peroxidases, which are important regulators of intracellular ROS levels and catabolism. Consequently, ROS are abundant in beta cells and cause oxidative damage.

**Figure 3 ijms-24-03082-f003:**
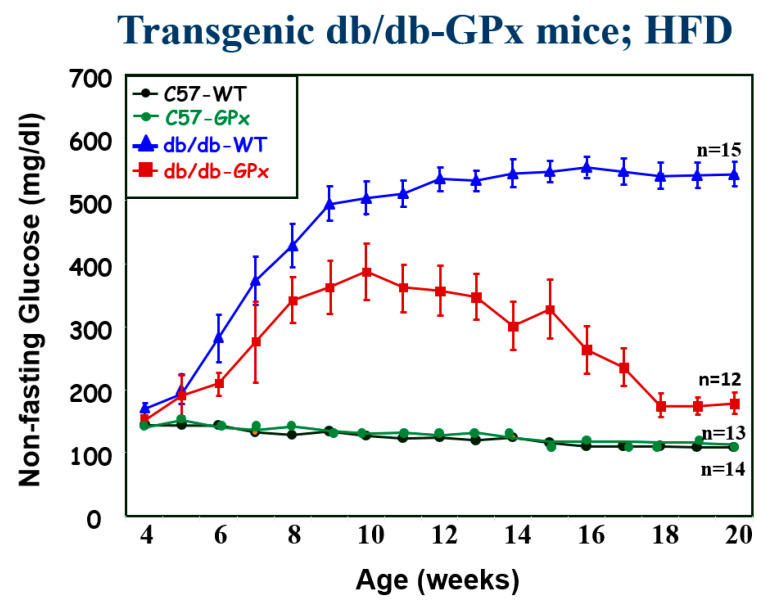
Beta cell-specific overexpression of glutathione peroxidase protects beta cells from the functional deterioration observed in wild-type *db/db-GPx(−)* mice fed a high-fat diet. Initially, blood glucose levels began to rise in both the C57 wild type and the *db*/*db* transgenic animals. However, by 10 weeks glucose levels began to decrease to levels lower than those in the transgenic animals, and thereafter returned to the non-hyperglycemic range. We speculate that this delay in glucose response may have been related to the fact that the GPx transgene construct included a glucose-sensitive insulin promoter that was activated by the establishment of hyperglycemia. Modified from Ref. [21].

**Figure 4 ijms-24-03082-f004:**
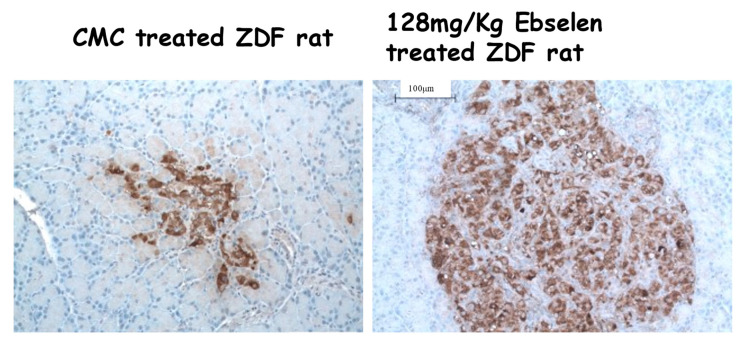
Comparison of islet morphology in a control (CMC) and an Ebselen-treated ZDF rat. Taken as a whole, the average total beta cell mass in the ebselen group doubled. Bar in left corner of right image indicates relative sizes of both images. Modified from Ref. [22].

**Figure 5 ijms-24-03082-f005:**
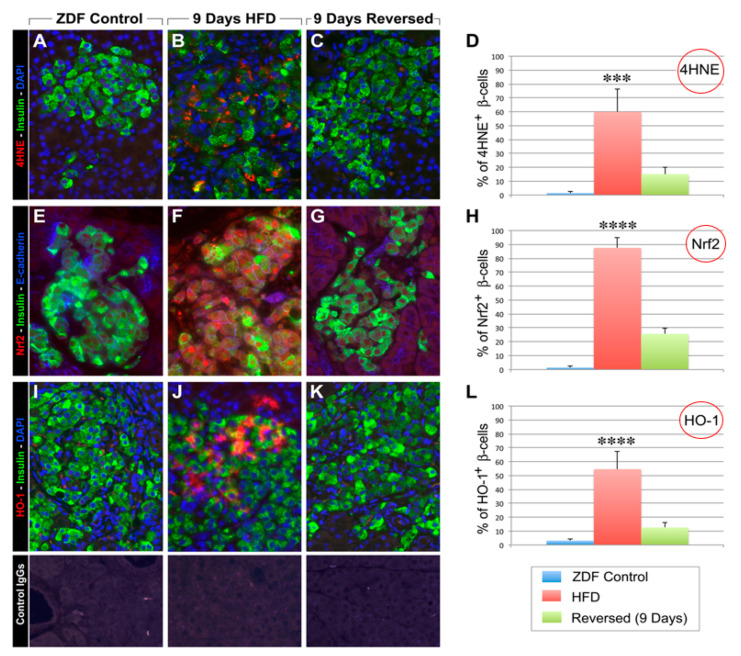
(**A**) Pancreatic sections were double labeled for insulin (green fluorescence) and 4-HNE (a marker for oxidative stress; red fluorescence). ZDF rats after 9 days of a high-fat diet showed intense cytoplasmic staining for 4-HNE in beta cells (**B**). This was reduced 2 weeks after a return to regular diets (**C**). Immunostaining for Nrf-2 (red fluorescence) revealed significant immunoreactivity both in the cytoplasm and the nucleus of beta cells ZDF rats fed high-fat diets for 9 days (**F**) when compared to ZDF controls (**E**). In contrast, 2 weeks after returning to regular diets the immunoreactivity for Nrf2 was dramatically reduced (**G**). Similarly, pancreatic sections stained for HO-1 (red fluorescence) showed increased cytoplasmic and nuclear localization of HO-1 (**J**) in beta cells (green fluorescence), which after 9 days of HFD was greatly diminished 2 weeks after return to regular diet (**K**). Specificity of detected immunoreactivities was validated by incubation of tissue sections with control IgGs from each species (lower panels). *Morphometric analysis of markers of oxidative stress.* To determine the presence of oxidative stress, pancreata from both control ZDF rats fed with a 17% fat diet and HFD-treated ZDF rats fed with a 48% fat diet were fixed overnight with 4% formalin. Fixed tissues were processed for paraffin embedding, and 4-μm sections were prepared and mounted on slides. Sections from both groups of animals were chosen at 25-μm intervals for immunolocalization of 4HNE, Nrf2, and HO-1, as well as insulin to identify β cells and E-cadherin to identify the pancreatic epithelium. Fifty sections per animal group were analyzed for morphometric measurements. Primary antibodies used for these experiments included anti-4HNE (Abcam, ab46545), anti-Nrf2 (Abcam, ab31163), and anti–HO-1 (Enzo Life Sciences), all used at 1:100 dilution and incubated overnight at 4 °C. Following reaction with fluorophore-conjugated secondary antibodies (Jackson ImmunoResearch) slides were mounted with DAPI containing mounting medium (Vector Laboratories) and viewed at a Nikon i90 microscope for image acquisition and analysis using NIS-Elements AR 3.2 (Nikon). *** *p* < 0.001, **** *p* < 0.0001. Modified from Ref. [23].

**Figure 6 ijms-24-03082-f006:**
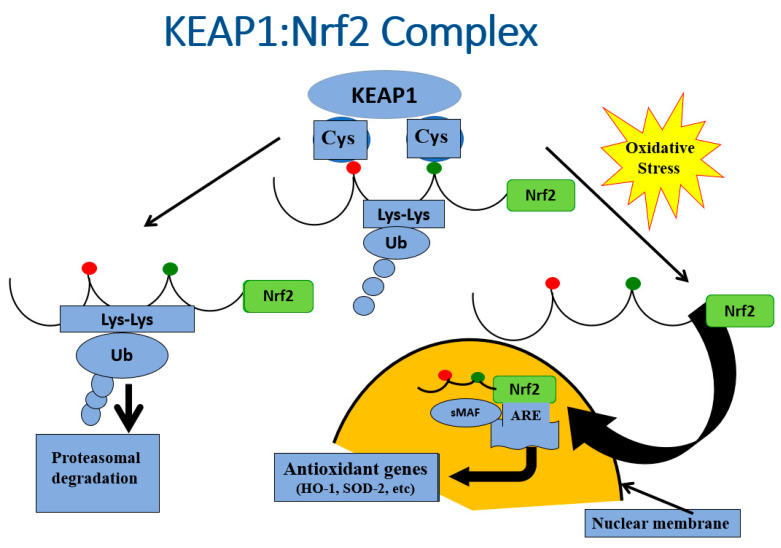
Illustration of the functional relationships between KEAP1 and Nrf2 under conditions of non-stress and oxidative stress. Under quiescent conditions, Nrf2 is bound by KEAP1 in the cytoplasm, which ushers Nrf2 to proteosomes for degradation. In the face of oxidative stress, Nrf2 is released from KEAP1 and enters the nucleus, where it serves as a key activator for the promoter of antioxidant genes.

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
