# Peer review of "Nrf2 and Antioxidant Response in Animal Models of Type 2 Diabetes"

_ijms, 2023, doi:10.3390/ijms24043082_

Round 1

Reviewer 1 Report

in this manuscript author examined the antioxidant capability of beta cells under  hyperglycemia of T2D. 

Although the manuscipt is quite interesting, it need important revisions. In fact, the main text of the manuscript must be significantly reorganized since abstract is included in the correspondence section and, keywords and introduction are missed.

Introduction: author should highlight the multifaceted role of NRF2/KEAP1 pathway. In fact, it deserves to be added that this pathway is also involved in preventing cancer onset (by increasing the antioxidant responce of normal cells, see PMID: 36335520), progression and in improving chemotherapy responce (as recently reviewed PMID: 35901941, 36289931, 35453348). This in an important point to add since it can highlight the intersing results obtained by the authors. 

Figure 4: scale bars must be formatted and located in the same format and position

Figure 6: Author must explain how immunofluorescence images have been quantified since HO-1, 4HNE+ and Nrf2+ cells were not counted in reference 38 mentioned by the author

Author Response

see attached pdf

Reviewer 2 Report

This perspective article by Robertson evaluates the success of endogenous Nrf2 and antioxidant gene expression to repair beta cells in Type 2 diabetes. The author did a wonderful job providing a brief background and summarizing their previous studies. However, I have following concerns:  

Major:  

  1. Articles just provides brief summaries of previous studies and personal perspective/opinion is missing. The manuscript will improve if you can include discussion/summary section where you describe how all your studies are flowing together and implications of your studies, GSH and Nrf2, in terms of Type 2 diabetes.  

Minor: 

  1. Keywords are missing. 
  2. Provide a brief description (maybe a couple of lines for each) of different sources of ROS and antioxidant regulatory response. 
  3. Rewrite lines 6 to 10 on page 2.  
  4. Page 2, line 13—by a high fat diet (see below) --- Please clarify which figures are referring to here because figure 1 is present below this text and doesn’t match the text. 
  5. References missing on first two paragraphs of Page 4.  
  6. In the first paragraph of page 4, the difference (you are mentioning) between T1D and T2D is not clear. Initial burst of insulin release upon glucose injection was totally missing in T2D patients and     T1D also included total disappearance of agonist-induced insulin responses. If both are true, then please clarify the difference you are trying to mention. 
  7. Page 4, paragraph 3, elaborate lines 6 to 8.  
  8. Page 4, please introduce troglitazone.  
  9. Figure 3 is not referred to anywhere in the manuscript.  
  10. Page 6, you are defining ZDF rats, but you have mentioned this model earlier as well. Therefore, it would be better to introduce ZDF rats earlier.  
  11. Page 7, Subheading V- Protection of beta cells-----db/db mice, please rewrite or elaborate lines 5 to 7. The message is not clear.  
  Overall, I would recommend to accept the manuscript with minor changes.   

Author Response

see attached pdf

Round 2

Reviewer 1 Report

the manuscript has been significantly improved and can be accepted in the present form